# Passport-aware Normalization for Deep Model Protection

**Jie Zhang**[1]*
zjzac@mail.ustc.edu.cn

**Dongdong Chen**[2][†]*
cddlyf@gmail.com

**Jing Liao**[3]
jingliao@cityu.edu.hk

**Weiming Zhang**[1]
zhangwm@ustc.edu.cn

**Gang Hua**[4]
ganghua@gmail.com

**Nenghai Yu**[1]
ynh@ustc.edu.cn

[1]University of Science and Technology of China    [2]Microsoft Cloud AI
[3]City University of Hong Kong    [4] Wormpex AI Research

## Abstract

Despite tremendous success in many application scenarios, deep learning faces serious intellectual property (IP) infringement threats. Considering the cost of designing and training a good model, infringements will significantly infringe the interests of the original model owner. Recently, many impressive works have emerged for deep model IP protection. However, they either are vulnerable to ambiguity attacks, or require changes in the target network structure by replacing its original normalization layers and hence cause significant performance drops. To this end, we propose a new passport-aware normalization formulation, which is generally applicable to most existing normalization layers and only needs to add another passport-aware branch for IP protection. This new branch is jointly trained with the target model but discarded in the inference stage. Therefore it causes no structure change in the target model. Only when the model IP is suspected to be stolen by someone, the private passport-aware branch is added back for ownership verification. Through extensive experiments, we verify its effectiveness in both image and 3D point recognition models. It is demonstrated to be robust not only to common attack techniques like fine-tuning and model compression, but also to ambiguity attacks. By further combining it with trigger-set based methods, both black-box and white-box verification can be achieved for enhanced security of deep learning models deployed in real systems.

## 1 Introduction

Deep learning has achieved huge success in broad artificial intelligent tasks, such as image recognition [1, 2, 3], object detection [4, 5, 6], and neural language processing [7, 8]. In order to obtain high-performance deep models, we often need to design a good network architecture, collect massive high-quality training dataset, and consume expensive computation resources. Therefore, these deep models are of great commercial value and may even be the core techniques for some companies. However, recent works [9, 10, 11] have shown that deep models are vulnerable to IP infringement. For example, the attackers can utilize transfer learning to adapt the target model to a new task by fine-tuning [12] or even get a new efficient model by model compression techniques [13]. All these attack methods will seriously infringe the interests of the original model owner.

In the past several years, deep model IP protection has drawn much attention from both academia and industry and many great works emerge. The main idea of these works is to add some special watermarks into the network weights [9, 14] or predictions [10, 15, 11, 16] while trying to maintain the original model performance. In detail, a weight regularizer is added into the objective loss function in [9, 14] so that the learned weights can follow some special distribution. In contrast, prediction watermarking [10, 15, 11] adds a special image trigger set into the training process, so that the learned network can classify them into some pre-defined labels. Despite the effectiveness in resisting the aforementioned IP attacks, a recent work [17] shows that these methods are all fragile to ambiguity attacks, i.e., the attackers can embed another watermark into the watermarked model for ownership claim, thus causing ambiguous forensics. To address this problem, a passport-based method is proposed by modulating the network performance based on the passport correctness. In other words, the target model can get good performance only when the correct passport is given and they use such "fidelity verification" to resist ambiguity attacks. However, since the passport learning and target model learning are coupled too tightly in [17], we find they have a major limitation: they need to change the network structure by replacing normalization layers, which may affect the original model performance significantly. Both of them are unfriendly and harmful to service quality for the end-users.

This paper shares a similar spirit in defending against ambiguity attacks, but targets at no network structure change and less performance drop. To this end, a new passport-aware normalization formulation is proposed. It is generally applicable to most popular normalization layers and only needs to add an extra passport-aware branch for IP protection. During training, some secret passports are pre-defined and these extra branches are jointly trained with the target model. After training, both these secret passports and new branches will be kept by the model owner for future ownership verification, and only the original target model is delivered to end-users to run the inference. Therefore, from the end-users' perspective, there is no network structure change. Moreover, since the normalization statistics (e.g., the running mean and variance of Batch Normalization) of the passport-aware branch are designed to be computed independently, less performance influence will be introduced to the target model.

When we suspect one model is illegally derived from the target model, we can add the private passport-aware branch back for ownership verification. More specifically, the target model performance will remain intact only when the correct passport is given, or seriously degrade for the forged passport. The effectiveness of our method is verified on both image and 3D point recognition models via comprehensive experiments, which demonstrate our method is not only robust to common removal attack techniques like fine-tuning and model compression but also to ambiguity attacks. By further combining it with trigger-set based watermarking schemes, we can achieve initial verification without the need of detailed model structure and weight access, which is known as the black-box verification.

To summarize, our main contributions are two-fold: 1) We propose a new and general passport-aware normalization formulation for deep model IP protection, which is compatible with most popular normalization layers. To the best of our knowledge, this is also the first passport-based method without the need of network structure change while achieving much less model performance degradation. 2) We have conducted extensive experiments on both image and 3D point recognition tasks with different network structures and normalization layers, which well demonstrate the effectiveness and robustness of our method against both removal attacks and ambiguity attacks.

## 2 Related Work

**Model IP protection.** Because of the underlying commercial value, IP protection for deep models has drawn increasing interests from both academia and industry. Inspired by traditional media IP protection techniques, many works [9, 14, 10, 15, 11, 16] have been proposed in the past several years. For example, [9] is possibly the first watermarking algorithm for DNNs, which attempts to embed bit watermark into the weights by adding an additional regularization term into the objective loss function. A similar weight watermarking idea is also adopted in [14] from the fingerprint perspective. Though these two methods are resilient to attacks such as fine-tuning and pruning, they need to access the target model structure and weights in a white-box way for forensics. To enable remote ownership verification in a black-box way, Adi *et al*. [10] add a special set of data into training and force the network to classify these data into pre-define labels. Following this idea, [15, 11] propose to utilize adversarial examples or watermarked images as triggers.

However, as shown in the latest work [17], all the above methods are shown to be fragile to ambiguity attacks. To address this limitation, Fan *et al.* [17] propose to add a passport layer into the target network and build the connection between the network performance and the passport correctness to resist ambiguity attacks. However, this method only works for some special normalization layers (e.g., group normalization[18]), thus the target network structure often needs to be changed to have the protection. For many tasks, this will incur significant performance drops. Our work is motivated by [17], but the newly proposed passport-aware normalization formulation is general to most existing normalization layers and decouples the passport-aware learning and passport-free learning. The following parts will show [17] can be seen as one special case of our general formulation and our method has better generalization ability and performance.

**Normalization Layers.** In the deep learning era, normalization layers play a crucial role, which can significantly ease the network training and boost the performance. For different tasks and training consideration, different types of normalization layers have been designed, such as Batch Normalization (BN) [19], Group Normalization (GN) [18], Instance Normalization (IN) [20] and Layer Normalization (LN) [21]. Generally, all these normalization layers follow a similar formulation, i.e., first normalize the input feature then denormalize back with an affine transformation:

$$\hat{x} = \gamma \frac{x - \mu(x)}{\sigma(x)} + \beta, \tag{1}$$

where $\mu, \sigma$ are the functions to calculate the mean and standard deviation (std) of $x$ respectively. They are also the key differences among different normalization layers. As the most widely used normalization layer, BN also differentiates from other normalization layers in the statistics usage. In details, GN,IN,LN calculate the mean/std statistics on-the-fly during inference but BN uses the training-stage moving averaged mean/std statistics. Therefore, accurate moving average mean/std statistics are especially important in BN to achieve good performance. In this paper, we only consider target models with normalization layers and propose a general normalization formulation tailored for IP protection so that it can generalize to different types of normalization.

# 3 Method

## 3.1 Problem Definition

To protect the IP of a target model $M$, we use a passport based ownership verification scheme to resist both removal attacks (e.g., fine-tuning, model compression) and ambiguity attacks. In detail, the target model performance is normal only when a correct passport is given, but deteriorates significantly for a forged passport, making the passport uniqueness a valid evidence for forensics. Moreover, we believe a good IP protection technique should satisfy two criteria: 1) It should be as general as possible and does not require any network structure change; 2) It should affect the original model performance as little as possible, otherwise it will hurt the original model competitiveness.

## 3.2 Passport-aware Normalization

To meet the above requirements, we give the key design motivations of our method here. First, to build the relationship between the model performance and the passport correctness, we leverage the denormalization step in Equation (1) and change the affine transformation parameters $\gamma, \beta$ to be the function of passport. In this way, if an incorrect passport is fed into the target model, the generated affine transform parameters will also be incorrect and make the model work abnormally. Besides, since the target model should not be changed when delivered to end-users, we also learn another set of affine transformation parameters $\gamma_0, \beta_0$ for the original model. Second, as described above, accurate moving average mean/std statistics are very crucial to BN's performance. However, we find the feature statistics learned by passport related $(\gamma, \beta)$ will be significantly different from that learned by passport-free $(\gamma_0, \beta_0)$, thus calculating the statistics together will seriously affect the original model performance. In view of that, we propose to calculate the statistics independently when introducing passport into the network training. Taking the above two motivations together, a

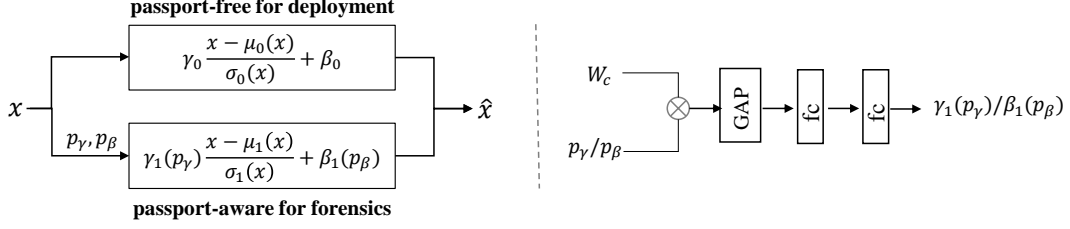

Figure 1: Illustration of the proposed passport-aware normalization. We add one independent passport-aware branch into existing popular normalization layers for IP protection, whose affine transform parameters $\gamma_1, \beta_1$ are designed to be relevant to both the precedent convolution weight $W_c$ and pre-defined passport $p_\gamma, p_\beta$.

new passport-aware normalization layer is proposed as shown in Figure 1. And its mathematical formulation is as follows:

$$
\hat{x} = \begin{cases} \gamma_0 \frac{x-\mu_0(x)}{\sigma_0(x)} + \beta_0 & \text{passport-free;} \\[2mm] \gamma_1(p_\gamma)\frac{x-\mu_1(x)}{\sigma_1(x)} + \beta_1(p_\beta) & \text{passport } p_\gamma, p_\beta. \end{cases} \tag{2}
$$

Comparing Equation (2) with Equation (1), it can be seen that the proposed passport-aware normalization only adds an extra passport-aware branch. For normalization after fully connected layers, this branch will calculate its own normalization statistics $\mu_1, \sigma_1$ and learn the affine transform parameters $\gamma, \beta$ based on the passport $p_\gamma, p_\beta$ respectively. For stronger ownership claim, we design the learnable $\gamma_1, \beta_1$ to be also relevant to the original model weight, i.e.,

$$
\begin{aligned}
\gamma_1(p_\gamma) &= w_2^T g(w_1^T(wp_\gamma)), \\
\beta_1(p_\gamma) &= w_2^T g(w_1^T(wp_\beta)), \\
where \quad wp_x &= GAP(W_c \otimes p_x)
\end{aligned} \tag{3}
$$

where $\otimes$ denotes the convolution operator and $W_c$ is the kernel weights of the precedent convolution layer. For the normalization layer after fully connected layers, $W_c \otimes p_x$ will be replaced by $W_c^T p_x$. $GAP$ is the global average pooling operator that converts the convoluted passport into a vector whose size is same as $\gamma_1, \beta_1$. $w_1, w_2$ are the weights of two fully connected layers (without bias term) to be learned, and $g$ is the non-linear activation function (Leakly ReLU used by default).

Need to note that, for some normalization layers like GN and IN, since their normalization statistics are calculated on-the-fly during the inference stage, $\mu_0, \sigma_0$ are exactly the same as $\mu_1, \sigma_1$. But for BN, $\mu_0, \sigma_0$ and $\mu_1, \sigma_1$ used in the inference stage are the moving average statistics calculated in the training stage from two branches, so their values are significantly different.

In order to train the target model with the proposed passport-aware normalization, we adopt a simple but effective alternating training strategy. Specifically, we will first pre-define the passport $p_\gamma, p_\beta$ for each normalization layer, then train the passport-free branch and passport-aware branch in an alternating way. Though the passport-aware and passport-free branch are trained alternately by default, experimental results show that they can also be trained simultaneously with similar performance.

**Relationship to [17].** To resist ambiguity attacks, Fan *et al*. added a passport layer into the target network, which follows the below formulation:

$$
\hat{x} = wp_\gamma * x_p + wp_\beta, \tag{4}
$$

where $x_p$ is the input of the passport layer. In their detailed implementation, they have to replace all the BN layers in the target model with GN layers and use a single $\mu, \sigma$ ($\mu_0 = \mu_1, \sigma_0 = \sigma_1$) to normalize $x_p$ without affine transformation before feeding into the passport layer. That is to say, they do not use the two-branch decoupled way for mean/std statistics calculation as our method. This makes their method not work for target networks with BN layers. Besides, the affine transformation parameters in their formulation are not learnable. This will also make the transformed features incompatible with the normal features without the passport and incur training interference and

performance drops. Mathematically, our proposed formulation is more general and Equation (4) can be seen as a special case of Equation (2) with one branch learning ($\mu_0 = \mu_1, \sigma_0 = \sigma_1$) and non-learnable $\gamma_1, \beta_1$.

**Ownership Verification.** As described above, the passport-aware branch along with the pre-defined passports will be both kept by the model owner as secrets, and only the passport-free branch will be delivered to end-users. Therefore, there is no structure change in the target model for end-users. When we suspect one model is illegally derived from the target model, we can start the official forensics procedure by law enforcement. In detail, we add the corresponding passport-aware branches and pre-defined passports back to the illegal model. If this illegal model has exactly the same functionality as the original target model, we can use the relationship between the model performance and the passport correctness as the forensics evidence. Because only if the correct passport is given, the target model can keep its original performance, otherwise it suffers from a significant performance drop, which is called "fidelity verification". Even if this illegal model is fine-tuned from the target model and has different functionality, we can leverage the passport signature as the forensics evidence, which is defined as:

$$b_{p_\gamma} = sign(wp_\gamma), \quad b_{p_\beta} = sign(wp_\beta), \tag{5}$$

where $sign(x)$ is the sign function whose value is 1 when $x > 0$ else 0. As Equation (5) is not that sensitive to the absolute value of $W_c$, even though the illegal model has some weights change or functionality change, the passport signature is still relatively robust.

Though the above verification scheme works well in official legal forensics, it requires access to the detailed model weights and cannot support block-box (remote) verification. For some application cases where the illegal model can be remotely tested (e.g., cloud API service), it would be great if some initial verification can be achieved before the legal process. To support it, we combine our passport based method with existing trigger-set-based IP protection methods. Specifically, besides the original training dataset $\{X_s, Y_s\}$, we add a special set of data $X_t$ with self-defined labels $Y_t$ into the network training. In this way, we can remotely call the service by feeding $X_t$ into the suspect model and utilize the prediction accuracy with respect to $Y_t$ for initial verification.

**Loss Function.** The objective loss function of our method mainly consists of three different parts: the task-related loss for the original target model, optional trigger-set based IP protection loss and passport signature regularization loss.

$$\mathcal{L}_{total} = \sum_{\{X_s, Y_s\}} L(M(x), y) + \lambda_1 \sum_{\{X_t, Y_t\}} L(M(x), y) + \lambda_2 \sum_{l=1}^{n} \sum_{i=1}^{C_l} \sum_{x \in \gamma, \beta} max(\alpha_0 - b_{p_x,i}^{gt,l} * wp_{x,i}^l, 0),$$
$$\tag{6}$$

where $L$ is the task-related loss function like the cross-entropy loss used in classification. $b_{p_x,i}^{gt,l} \in \{-1, 1\}$ is $i$th bit of the pre-defined ground truth passport signature at layer $l$ and $\alpha_0$ is a small positive constant that encourages $wp_{x,i}^l$ to be larger than $\alpha_0$. $n$ is the total number of passport-aware normalization layers and $C_l$ is the corresponding feature channel number.

# 4 Experiments

To demonstrate the effectiveness and superiority of our method, we apply the proposed passport-aware normalization on two representative tasks: image classification on the CIFAR10 and CIFAR100 [22] dataset, and 3D point recognition on the ModelNet [23] and ShapeNet [24] dataset. For image classification, we follow the typical setting and use the well-known AlexNet and ResNet-18 structure, while for 3D point recognition, the popular point recognition network PointNet [25] is adopted. As for trigger-set, we adopt a similar setting mentioned in [10]. That is, we use about 100 images/points not belonging to the target dataset as the trigger set for all tasks. In this section, we first explain why the target network should not be changed and provide a comparison of the performance influence on the target model. Then we demonstrate the robustness to both removal attacks and ambiguity attacks. Finally, ablation analysis is given to justify the importance of our design. Since we aim at resisting both removal attack and fine-tune attack, we only consider the passport-based method [17] as the baseline and report their results by running the code they publicly released. We shall point out that we utilize passport-aware normalization on all layers of adopted networks and trigger appended into training by default.

Table 1: The performance comparison by using different normalization layers on three different networks, which shows replacing BN with GN will incur substantial performance drops.

| Normalization | AlexNet | | ResNet-18 | | PointNet | |
|---|---|---|---|---|---|---|
| | CIFAR10 | CIFAR100 | CIFAR10 | CIFAR100 | ModelNet | ShapeNet |
| BN | 91.20 | 68.57 | 95.15 | 76.60 | 90.20 | 99.57 |
| GN | 90.43 | 65.10 | 93.67 | 72.43 | 88.87 | 99.41 |

Table 2: Model performance comparison for deployment/verification. Obviously, our method can achieve much better accuracy than the baseline [17] for both deployment and verification no matter BN or GN is used.

| Accuracy | | Baseline [17](BN) | Baseline[17] (GN) | Ours(BN) | Ours(GN) |
|---|---|---|---|---|---|
| AlexNet | CIFAR10 | 71.57 / 76.12 | 87.73 / 86.31 | **90.00 / 89.97** | 89.01 / 88.37 |
| | CIFAR100 | 32.01 / 17.99 | 61.08 / 59.80 | **66.47 / 63.95** | 63.01 / 60.43 |
| ResNet-18 | CIFAR10 | 21.99 / 10.60 | 92.46 / 91.46 | **94.25 / 93.66** | 92.82 / 91.60 |
| | CIFAR100 | 1.90 / 3.10 | 67.19 / 64.51 | **74.40 / 73.54** | 69.99 / 66.48 |
| PointNet | ModelNet | 81.00 / 4.03 | 86.94 / 86.77 | **90.20 / 89.35** | 88.19 / 87.82 |
| | ShapeNet | 96.05 / 53.21 | 99.03 / **98.97** | **99.31 / 99.47** | 99.14 / 98.93 |

## 4.1 Generalization ability and performance comparison

BN is very important and widely used in modern image and 3D recognition networks, including AlexNet, ResNet-18 and PointNet. As an alternative of BN, GN is designed to address the learning problem caused by a small batch size. On the other hand, since GN computes the mean/std statistics on-the-fly during inference, it does not need training-stage moving average statistics. However, GN still cannot match BN's performance in many recognition tasks. As shown in Table 1, replacing BN with GN in AlexNet, ResNet-18 and PointNet will all incur substantial performance drops. Especially for ResNet18 on the CIFAR100 dataset, there is even $4.2$ points drop. However, in the baseline method [17], the network structure must be changed by replacing all BN layers with GN, otherwise, the target model will not work when the passport is involved in the training.

In Table 2, we provide a detailed comparison for the model performance when removing the passport-related parts for end-users deployment and adding them back for ownership verification (correct passport is given). It can be seen that when using BN in [17], both the deployment and verification accuracy will significantly drop. Taking ResNet-18 as example, the deployment top-1 accuracy degrades from $95.15\%$ to $21.99\%$ on CIFAR10 and from $76.6\%$ to $1.90\%$ on CIFAR100. In contrast, with the newly proposed passport-aware normalization, our method works well for BN with $94.25\%$ and $74.40\%$ top-1 accuracy respectively. Even with GN, our method is still better than [17] with less performance drop when involving the passport into training for IP protection. To summarize, compared to [17], our method not only has stronger generalization ability without the need of network structure change but also maintains the original model performance better. Compared to the original model performance without IP protection in Table 1, we observe that a slight performance drop still exists after involving passport into training. The following analysis will show this can be rescued by only adding passport-aware branches to some but not all the normalization layers.

To further compare with transformation based normalization method, we tried the famous conditional normalization layer SPADE in [26]. Considering the conditional input in our task is the one-dimension transformed passport $wp_\gamma, wp_\beta$, we replace the convolution layer used in SPADE with FC layer. Then it can be viewed as a special case (i.e., only the nonlinear transform in Equation (3)) of our method without the decoupled design in Equation (2).

## 4.2 Robustness Comparison

To demonstrate the robustness of our method, we follow a similar setting as [17] and consider both removal attacks (cross-dataset fine-tuning, model compression) and ambiguity attacks.

**Fine-tuning.** For fine-tuning attacks, we first train the target model with the passport-aware branch on one dataset then fine-tune the trained model on a new dataset without it. As shown in Table 3, two different settings are tried, i.e., CIFAR10→CIFAR100 and CIFAR10→Caltech-101. Since the

Table 3: The performance (in-bracket) and signature detection accuracy (out-bracket) after fine-tuning. Both the original model and baseline method are displayed for comparison.

| Method | CIFAR10 → CIFAR100 | | CIFAR10 → Caltech-101 | |
|---|---|---|---|---|
| Original | AlexNet (66.16) | ResNet-18 (75.71) | AlexNet (74.25) | ResNet-18 (79.66) |
| Baseline [17] | 98.70 (61.79) | 98.57 (71.06) | 99.92 (72.43) | 99.98 (72.88) |
| Ours | 98.88 (63.74) | 98.26 (73.92) | 100.00 (74.75) | 99.97 (77.91) |

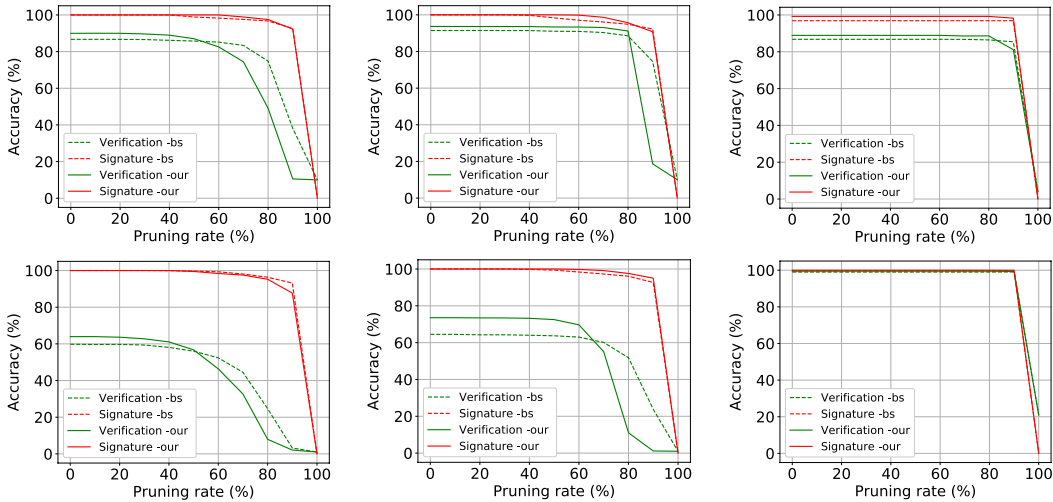

(a) AlexNet (top: CIFAR10; bottom: CIFAR100)

(b) ResNet18 (top: CIFAR10; bottom: CIFAR100)

(c) PointNet (top: ModelNet; bottom: ShapeNet)

Figure 2: The performance of model for verification and signature detection rate after model pruning. The signature maintains the robustness (more than 90% detection rate) even with a fierce pruning rate (90%) in all cases.

fine-tuned model is on a new task, we leverage the passport signature defined in Equation (5) for verification. For one specific signature bit sequence, it will be seen successfully detected only if all the binary bits are exactly matched. Obviously, the signature can be detected with more than 98% accuracy in all cases like the baseline method [17]. As explained before, this is mainly because using the sign function makes the verification insensitive to the detailed network weight value change.

**Model pruning.** For model compression attacks, we adopt the same class-blind pruning scheme [13] used in [17], which is designed to reduce the redundant parameters within a model without affecting its performance too much. In Figure 2, we show comparison results between our method and the baseline method [17] on the accuracy of the verification model with correct passport and the accuracy of the signature detection. It can be seen that even at 90% pruning rate, more than 90% signature can be detected in all cases.

### 4.3 Robustness against Ambiguity attacks

As for ambiguity attacks, we consider two type of adversarial settings from the perspective of attackers. For ambiguity attacks I, the attacker only has access to the model and utilizes reverse engineering to forge a fake passport from random values to achieve comparable model performance by gradient descent. In Table 4, we provide both the model performance of the initial random passport and the optimized passport. It shows that this type of attack cannot succeed for either our method or the baseline method with bad model performance. Besides, because of our two-branch and learnable design, the reverse-engineered performance of our method is even worse. In the case of ambiguity attacks II, we assume the attacker illegally obtained the original passport ($p_\gamma$ and $p_\beta$) and try to generate fake signatures by both guaranteeing the accuracy and increasing the dissimilarity between the fake signatures and original signatures by flipping the sign of $wp_\gamma$ and $wp_\beta$. Compared with the baseline in [17], since we also leverage non-trivial learnable $\gamma_1, \beta_1$ mentioned in Equation (3)

Table 4: The performance of passport model with fake passport ( Ambiguity Attack I ). All results are the average accuracy calculated from 10 fake passports (random initial / the optimized).

| I | AlexNet | | ResNet-18 | | PointNet | |
|---|---|---|---|---|---|---|
| | CIFAR10 | CIFAR100 | CIFAR10 | CIFAR100 | ModelNet | ShapeNet |
| Baseline | 9.99/62.88 | 1.01/9.27 | 10.08/54.37 | 1.00/11.50 | 4.03/19.90 | 31.46/77.20 |
| Ours | 10.07/44.00 | 1.00/4.88 | 9.83/41.23 | 1.00/7.46 | 4.03/6.63 | 31.46/37.46 |

Table 5: The performance of passport model with fake passport ( Ambiguity Attack II ) under different percentage of flipped sign. We take the results of ResNet-18 on CIFAR100 as example.

| II | 10% | 20% | 30% | 40% | 50% | 60% | 70% | 80% | 90% | 100% |
|---|---|---|---|---|---|---|---|---|---|---|
| Baseline [17] | 55.73 | 47.73 | 14.51 | 5.82 | 6.93 | 8.78 | 6.82 | 8.04 | 5.58 | 5.28 |
| Ours | 6.21 | 6.10 | 4.76 | 5.08 | 5.49 | 5.45 | 5.18 | 6.21 | 5.94 | 5.18 |

as an additional protective barrier against attack, our method is much harder to be attacked. As shown in Table 5, the proposed method is more robust than the baseline method [17]. Specifically, the adversary just obtains 6.21% accuracy even when only 10% of signatures are modified, which is beneficial for forensics.

## 4.4 Ablation Analysis

**Design importance.** In our method, we append one independent branch for passport-aware normalization (i.e., $\mu_0 \neq \mu_1, \sigma_0 \neq \sigma_1$ for BN) and design a learnable affine transform $\gamma_1, \beta_1$ to guarantee the performance of the passport model while preserving the original network structures. In this experiment, we conduct ablation analysis with three combinations. (A): No independent branch for passport-aware normalization and non-learnable $\gamma_1, \beta_1$ as [17]; (B) Independent branch for passport-aware normalization but with non-learnable $\gamma_1, \beta_1$; (C) Independent branch for passport-aware normalization with learnable $\gamma_1, \beta_1$. From Table 6, we observe that adding an extra branch for passport-aware normalization is very important and learnable affine transformation $\gamma_1, \beta_1$ can also bring extra performance gain. The train-val convergence curve of these three configurations are also shown in Figure 3. The independent branch for passport-aware normalization significantly reduces the gap between training and inference performance while the learnable affine transformation makes the original task more compatible with the passport related training.

**Influence of passport layer number.** In our default setting, we use the passport-aware normalization formulation in all the normalization layers. In other words, we can verify the ownership on all these layers. However, as shown in Table 2, involving passport into the training will affect the original model performance. In this experiment, we take the ResNet-18 on CIFAR100 as an example and conduct an experiment by only applying the passport-aware normalization into the last-three normalizaiton layers. It is shown that the original deployment accuracy will increase from 74.40% to 76.22%, which is very close to the original model performance 76.60% without passport training. Therefore, in real application systems, we can adapt the passport layer number to achieve a better balance between robustness and performance.

## 4.5 Training cost

Though the passport-aware and passport-free branch are trained alternatively by default, experimental results show that they can also be trained simultaneously with similar performance. For the training cost, it indeed depends on the ratio of the passport-aware branch activated in every training epoch. We further replace the default ratio 50% by a lower ratio 10%. Under this setting, take PointNet on ShapeNet dataset for example, the verification accuracy is almost unchanged (from 99.47% to 99.31%) while the model deployment performance is even slightly better (from 99.31% to 99.36%). More importantly, this will not introduce any extra cost for deployment.

## 5 Conclusion

With the huge success of deep learning, IP protection for deep models becomes more and more important and necessary. Though many works have been proposed along this direction, we find

Table 6: The deployment/verification performance of three different configurations about whether using independent branch for normalization (C1) and learnable affine transformation parameters(C2). A: C1-No, C2-No, B: C1-Yes, C2-No, C: C1-Yes, C2-Yes.

|   | AlexNet | | ResNet-18 | | PointNet | |
|---|---------|---------|---------|---------|---------|---------|
|   | CIFAR10 | CIFAR100 | CIFAR10 | CIFAR100 | ModelNet | ShapeNet |
| A | 70.07/13.34 | 30.94/1.01 | 17.46/13.74 | 2.05/2.30 | 81.00/4.03 | 96.05/31.46 |
| B | 89.54/88.77 | 65.69/63.82 | 94.04/93.56 | 73.84/72.49 | 89.72/89.11 | 99.30/99.30 |
| C | **90.00/89.97** | **66.47/63.95** | **94.25/93.66** | **74.40/73.54** | **90.20/89.35** | **99.31/99.47** |

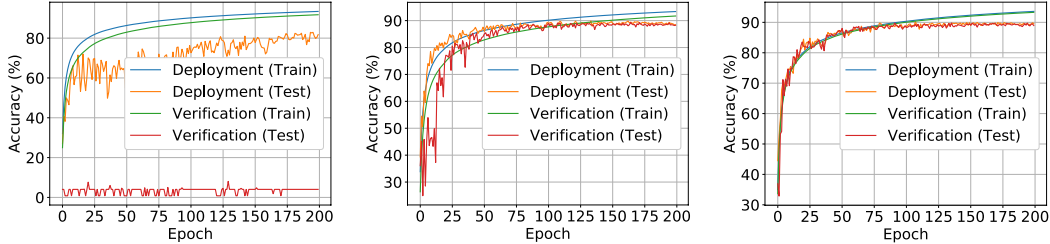

Figure 3: The train-val convergence curve for the A,B,C configuration shown in Table 6

they either are vulnerable to ambiguity attacks, or require changes in the target model structure and incur significant performance drops. To address these limitations, we propose a new passport-aware normalization formulation. It is general to most existing deep networks equipped with normalization layers and only needs to add an extra passport-aware branch into the normalization layers. Extensive experiments have been conducted on image and 3D point recognition models, which show the strong robustness of our method with less performance influence on the target model.

## Broader Impact

Though deep learning evolves very fast in these years, IP protection for deep models is seriously under-researched. In this work, we mainly aim to propose a general technique for deep model IP protection. It will help both academia and industry to protect their interests from illegal distribution or usage. We hope it can inspire more works along this important direction.

## Acknowledgments

This work was supported in part by the NSFC Grant U1636201, 62002334 and 62072421, Exploration Fund Project of University of Science and Technology of China under Grant YD3480002001, and by Fundamental Research Funds for the Central Universities under Grant WK2100000011, and Hong Kong ECS grant 21209119, Hong Kong UGC. Gang Hua is partially supported by National Key R&D Program of China Grant 2018AAA0101400 and NSFC Grant 61629301. And Jie Zhang is partially supported by the Fundamental Research Funds for the Central Universities WK5290000001.

## Footnotes

*Equal Contribution, † Dongdong Chen is the corresponding Author

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
