[Reviews · NeurIPS 2020]

Review 1

Summary and Contributions: The paper encode the passport into the affine transformation parameters in Batch Normalization Layer for deep model IP protection. The method is generally applicable to most existing normalization layers and only needs to add another passport-aware branch for IP protection.

Strengths: (1) The idea of independent branch for passport-free for deployment and passport-aware for verification is novel, only the passport-free branch will be delivered to end-users to prevent malicious attack and there is no structure change in the target model for end-users. (2) Compare to [17], Since the method use a separate branch for passport-awareness, it does not need to modify all the BN layers in the target model as [17] did

Weaknesses: (1) The training cost is a least 2x time since two identical sized branch need to be trained.

Correctness: The claims and methods are correct.

Clarity: The paper is well-written and easy to understand

Relation to Prior Work: Yes, they compare the Relationship to [17], another BN based passport method.

Reproducibility: No

Additional Feedback:


Review 2

Summary and Contributions: This work deals with establishing ownership of a DNN, which is an important problem given resources and IP involved in training accurate models. The authors propose a passport-aware scheme by batch normalization which is implemented by an independent branch and learnable shift parameters. Such a design can improve the performance of the previous method. Extensive experiments demonstrate the effectiveness of the proposed method.

Strengths: The paper is well-organized and easy to follow. The method is new and differs from previous contributions. Extensive experiments are conducted to verify the effectiveness of the proposed method.

Weaknesses: The design of passport-aware branch seems to be similar to the Squeeze-and Excitation block [1] that is a popular module in network architecture area. Please provide any intuitions why the authors choose the transformation like in eq3. When comparing eq3 with eq4, the improvement of passport-aware normalization over eq4 seems a bit incremental. Moreover, the proposed scheme also introduces extra complexity in model parameters and network training. [1] Jie Hu, et al. Squeeze-and Excitation Networks, CVPR2018.

Correctness: The technical details of the article are basically correct. The motivation is valid.

Clarity: The writing quality is good. The authors clearly present the contributions of the paper.

Relation to Prior Work: To my best of knowledge, the related work section sec.3 is complete.

Reproducibility: Yes

Additional Feedback: There seen to be many normalization methods focusing on transformation on gamma and beta. It would be great if comparisons are provided.


Review 3

Summary and Contributions: This paper proposed a passport-aware normalization design by adding a new normalization branch beside the original norm. layer for IP protection. This work is an extension of [17] “Rethinking deep neural network ownership verification: Embedding passports to defeat ambiguity attacks”, by addressing the network structure change and model performance drop issues.

Strengths: Contributions clearly stated and validated.

Weaknesses: Some concerns: 1. One big concern is, the proposed method is combined with existing trigger-set-based IP protection methods to support the black-box verification, however, if the existing trigger-set-based method is strong enough to verify the suspect model do we need an extra step (i.e., the proposed method) to confirm? Or is this two-step verification necessary? I think the authors should provide some discussions or data to claim it. 2. The training process of the passport-free branch and the passport-aware branch is in an alternative way. What's the training cost? Why the authors use this training strategy, rather than train them simultaneously? More details are appreciated. 3. The details of the trigger-set-based method are missing. What existing method did the author use? How many special sets of data are used to identify a suspect model? Will this impact the DNN performance? Could the authors provide some detailed evaluation results of this part?

Correctness: The proposed method is clearly claimed.

Clarity: This paper is well written and organized.

Relation to Prior Work: The auhtors clearly discussed the relationship between the proposed method and [17], i.e. [17] is one special case of the proposed method.

Reproducibility: No

Additional Feedback:


Review 4

Summary and Contributions: This paper considers the problem of intellectual property protection for a learned deep model. It extends previous work [17] which introduced the notion of a passport layer, whereby the performance of the network would be significantly deteriorated unless the genuine passport was supplied. However, the introduction of the passport layer made it not possible to use batch-normalization, and additionally degraded overall performance of the network somewhat. This paper modifies the passport layer formulation, allowing for the use of batch-normalization, and allowing for learnable parameters in the passport layer.

Strengths: As mentioned in the Broader Impact statement, IP protection for deep models seems like an important but currently under-researched area given the state of deep learning and usage in commercial products, so this paper could have wide relevance. It addresses a specific issue with the existing method of [17] and leads to a small but consistent increase in performance.

Weaknesses: The proposed method is somewhat incremental over the method of [17]. In order to produce a network for deployment (that does not require the passport), [17] uses multi-task learning to optimize performance of the network when the passport layers are used as well as when they are skipped. This paper essentially does the same thing, except the version where the passport layers are skipped is now a batch or group normalization layer, with normalization statistics decoupled from the passport layer, and the passport layer can also contain learnable parameters. However, the ablation analysis presented in the paper suggests that the learnable affine transformation parameters give only a very small improvement.

Correctness: From my understanding, yes.

Clarity: Overall yes, with some typos. However, the section describing the training on page 4, line 147-150, could be improved. Specifically, there is a difference between "alternative" - meaning "different from the usual standard", and "alternating" - meaning "switching between two modes/paths". I assume the authors meant "alternating". Beyond this change, it might be helpful to be explicit - the training consists of one pass/update with one branch, and then one pass/update with the other?

Relation to Prior Work: yes

Reproducibility: Yes

Additional Feedback: update after reading author feedback: after reading the author response, although there are some empirical improvements, I still have the above concerns with novelty, and so maintain my original rating.

[Author Response · NeurIPS 2020]

Thanks for the valuable comments. We are happy that most reviewers recognized the importance and novelty of our
work. Below we clarify each question and we hope reviewers can raise their scores based on the responses.

**[R1 & R3: Reproducibility]** In Section 4 (L194 $\sim$ L205), we have provided the detailed experiment settings. We do
not use any tricks but follow the classic settings for image classification and 3D recognition. Our idea is very easy to
implement, we will release our code immediately upon acceptance.

**[R1 & R3: Training cost and training strategy.]** Though the passport-aware and passport-free branch are trained
alternatively by default, experimental results show that they ***can also be trained simultaneously*** with similar perfor-
mance. For the training cost, it indeed depends on the ratio of the passport-aware branch activated in every training
epoch. The default ratio value is 50%, i.e., train 1 iteration passport-aware branch after training every 1 iteration
passport-free branch. In this case, the theoretical computation cost will be 2x. However, ***we find it feasible to use a***
***lower ratio for the passport-aware branch with very comparable performance***. For example, when the ratio is 10%,
i.e.,train 1-iteration passport-aware branch after training every 9 iterations passport-free branch, its extra computation
cost is only about 10%. Under this setting, take PointNet on ShapeNet dataset for example, the verification accuracy is
almost unchanged (from 99.47% to 99.31%) while the model deployment performance is even slightly better (from
99.31% to 99.36%). ***More importantly, this will not introduce any extra cost for deployment***.

**[R2:Motivation of the transformation in eq3?]** Though our transformation formulation looks similar to that in
SE, ***the underlying consideration is totally different***. In SE, the transformation is to learn a channel-wise attention
to enhance the feature representation. But in our method, our motivations are from another two aspects: 1) The
input of passport-aware transformation $wp_\gamma, wp_\beta$ ($wp_k = W_c \otimes p_k$) share the same convolution kernel $W_c$ with the
convolutional feature $x$, considering the physical meaning and value difference between $p_\gamma, p_\beta$ and $x$, the non-linear
transformation is used to ***remap the*** $wp_\gamma, wp_\beta$ ***to meaningful transformation parameters*** $\gamma_1, \beta_1$, we find it can improve
the training stability and performance (Table 6 and Figure 3). 2) Using this non-linear transformation ***can help increase***
***the difficulty of passport reverse-engineering***, thus enhancing the robustness (Table 4,5).

**[R2: Difference between (eq2,eq3) *vs* eq4, extra complexity?]** Compared to the baseline eq4, our formulation (eq2,
eq3) innovatively decouples the passport-free and passport-aware branch and uses an extra non-linear transformation,
which are both the keys of our stronger generalization ability and performance. Compared to baseline [17], we find the
empirical training time is almost the same. More importantly, it does not introduce any extra parameter or computation
into the deployed model but only during forensics and training.

**[R2: Comparison with transformation based normalization methods.]** Thanks for your suggestion. We tried the
famous conditional normalization layer SPADE in "Semantic Image Synthesis with Spatially-Adaptive Normalization".
Considering the conditional input in our task is the one-dimension transformed passport $wp_\gamma, wp_\beta$, we replace the
convolution layer used in SPADE with fc layer. Then it can be viewed as a special case (i.e., only the nonlinear transform
in eq3) of our method without the decoupled design in eq2. Like [17], it needs to replace BN with GN, otherwise it will
produce very bad results (e.g., 21.99/1.90 for ResNet-18 on CIFAR10/CIFAR 100 (ours 94.25/74.40)) .

**[R3: Is two-step verification necessary?]** Yes. As described in **L36-42, L85-88 and [17]**, though trigger-set-based
methods support black-box verification, they are fragile to ambiguity attacks. In details, the attacker can forge another set
of trigger images to make the ownership claim ambiguous. In contrast, our passport based method can resist ambiguity
attack but cannot support black-box (remote) verification (**L178-182**). ***Therefore, using the two-step verification can***
***combine the advantages of these two methods, i.e., not only support remote verification but resist ambiguity attack***.

**R3: Trigger-set-based details.** We adopt a similar setting as the trigger-set based method [10]. We use about 100
images/points not belonging to the target dataset as the trigger set for all tasks. Empirically, we find adding such trigger
sets into training only slightly affects the target model performance. For instance, with ResNet-18 on the CIFAR10, the
model performance only drops slightly to 94.25% from 94.36%. We will add such analysis to the final version.

**[R3:Broader impact.]** As R5 and L293-297 described, IP protection for deep models is an important but currently
under-researched area. Our method can help protect deep business models from illegal distribution or usage.

**[R5: Key differences with the baseline [17]].** As described in section "relationship to [17]" (L151-162), the formu-
lation (Eq4) of [17] is a special case of our formulation (Eq2). Compared to [17], we have two key differences: 1)
decoupling passport-aware branch from passport-free branch; 2) learnable non-linear affine transformation. We empha-
size these two points are both important and non-trivial. The former point can significantly boost the generalization
ability for passport-based IP protection methods without any architecture change (e.g., BN to GN). The latter point not
only improves training stability and performance (shown in Figure 3 and Table 6), but also significantly enhances the
robustness against ambiguity attack (shown in Table 4, 5) compared with the non-learnable parameters used in [17].

**[R5: Typos.]** Thanks, we will improve it in the final version.

[Meta-Review · NeurIPS 2020]

The paper proposes a method for intellectual property protection for a learned deep model. The problem is highly relevant and understudied. While the proposed method was considered too incremental with added complexity compared to previous work, there was enough new perspectives and ideas to warrant acceptance.